# Activation of the Anaphase Promoting Complex Reverses Multiple Drug Resistant Cancer in a Canine Model of Multiple Drug Resistant Lymphoma

**DOI:** 10.3390/cancers14174215

**Published:** 2022-08-30

**Authors:** Terra G. Arnason, Valerie MacDonald-Dickinson, Matthew Casey Gaunt, Gerald F. Davies, Liubov Lobanova, Brett Trost, Zoe E. Gillespie, Matthew Waldner, Paige Baldwin, Devon Borrowman, Hailey Marwood, Frederick S. Vizeacoumar, Franco J. Vizeacoumar, Christopher H. Eskiw, Anthony Kusalik, Troy A. A. Harkness

**Affiliations:** 1Division of Endocrinology and Metabolism, Department of Medicine, Saskatoon, SK S7N 0W8, Canada; 2Department of Anatomy and Cell Biology, Saskatoon, SK S7N 5E5, Canada; 3Department of Anatomy, Physiology and Pharmacology, Saskatoon, SK S7N 5E5, Canada; 4Department of Small Animal Clinical Sciences, Western College of Veterinary Medicine, Saskatoon, SK S7N 5B4, Canada; 5Department of Biochemistry, Microbiology and Immunology, Saskatoon, SK S7N 5E5, Canada; 6Department of Computer Science, Saskatoon, SK S7N 5C9, Canada; 7Department of Food and Bioproduct Sciences, Saskatoon, SK S7N 5A8, Canada; 8Department of Pathology and Laboratory Medicine, University of Saskatchewan, Saskatoon, SK S7N 5E5, Canada

**Keywords:** multiple drug resistant cancer, Anaphase Promoting Complex, canine lymphoma, cell culture, metformin

## Abstract

**Simple Summary:**

Multiple drug resistant cancers develop all too soon in patients who received successful cancer treatment. A lack of treatment options often leaves palliative care as the last resort. We tested whether the insulin sensitizer, metformin, known to have anti-cancer activity, could impact canines with drug resistant lymphoma when added to chemotherapy. All canines in the study expressed protein markers of drug resistance and within weeks of receiving metformin, the markers were decreased. A microarray was performed, and from four canines assessed, a common set of 290 elevated genes were discovered in tumor cells compared to control cells. This cluster was enriched with genes that stall the cell cycle, with a large component representing substrates of the Anaphase Promoting Complex (APC), which degrades proteins. One canine entered partial remission. RNAs from this canine showed that APC substrates were decreased during remission and elevated again during relapse, suggesting that the APC was impaired in drug resistant canines and restored when remission occurred. We validated our results in cell lines using APC inhibitors and activators. We conclude that the APC may be a vital guardian of the genome and could delay the onset of multiple drug resistance when activated.

**Abstract:**

Like humans, canine lymphomas are treated by chemotherapy cocktails and frequently develop multiple drug resistance (MDR). Their shortened clinical timelines and tumor accessibility make canines excellent models to study MDR mechanisms. Insulin-sensitizers have been shown to reduce the incidence of cancer in humans prescribed them, and we previously demonstrated that they also reverse and delay MDR development in vitro. Here, we treated canines with MDR lymphoma with metformin to assess clinical and tumoral responses, including changes in MDR biomarkers, and used mRNA microarrays to determine differential gene expression. Metformin reduced MDR protein markers in all canines in the study. Microarrays performed on mRNAs gathered through longitudinal tumor sampling identified a 290 gene set that was enriched in Anaphase Promoting Complex (APC) substrates and additional mRNAs associated with slowed mitotic progression in MDR samples compared to skin controls. mRNAs from a canine that went into remission showed that APC substrate mRNAs were decreased, indicating that the APC was activated during remission. In vitro validation using canine lymphoma cells selected for resistance to chemotherapeutic drugs confirmed that APC activation restored MDR chemosensitivity, and that APC activity was reduced in MDR cells. This supports the idea that rapidly pushing MDR cells that harbor high loads of chromosome instability through mitosis, by activating the APC, contributes to improved survival and disease-free duration.

## 1. Introduction

It is estimated that one in two people will develop cancer in their lifetimes, a staggering statistic to consider. Advances in surgical techniques, cancer-specific therapies, and small molecule inhibitors have contributed to survival and disease-free duration [1]. However, the development of multiple drug resistance (MDR) to first-line chemotherapeutic agents leaves few, if any, proven treatment options, and may signal a shift to palliative support rather than active therapy [2,3]. MDR can be inherent (prior to treatment) or acquired at any time after initial treatment responsiveness, with recrudescence as much as 20 years later [4,5]. Malignancies of the breast, blood, lung, and colon are particularly known for their relatively high rates of treatment resistance and acquired MDR. New treatment options are required for MDR cancers.

To identify the underlying cellular pathways involved in MDR development, and to identify novel agents that will promote reversal or prevention of MDR cancers, we previously applied in vitro MDR cell culture models of a variety of human cancers to identify molecular biomarkers and their post-translational modifications that correlate with both entry to, and exit from, an MDR state [6,7,8,9,10]. Simultaneously, we evaluated the utility of several insulin-sensitizing oral agents with anti-cancer activity that returns MDR cells to treatment-sensitive states [7,8,10]. Metformin, a drug used to treat type 2 diabetes globally, as well as a number of other diseases [11], can slow the growth of multiple human cancer cells in vitro [12]. Molecular events impacted by metformin include AMPK activation and TOR inhibition [13,14]. Meta-analyses have also shown that in many cases, type 2 diabetics taking metformin have a lower incidence of many cancers compared to patients not taking the drug [15,16]. Metformin appears to induce cell death in cancer cells by increasing the activity of cell death mechanisms, particularly apoptosis [17]. Furthermore, metformin stimulates the immune system, which plays a critical role in driving cell death.

Recent clinical trials have focused on whether metformin provides healthspan benefits in disease free individuals. The Metformin in Longevity Study (MILES), originally conducted on 14 elderly individuals with compromised glucose tolerance, demonstrated that metformin impacts many anti-aging transcriptional programs, such as metabolic pathways, extracellular matrix remodeling, and DNA mismatch repair [18]. A number of other trials are currently underway, including Targeting Aging with Metformin (TAME), which is a large multicenter trial that plans to involve 14 research centers and enroll 3000 non-diabetic individuals aged 65–80 who will be given metformin for 6 years with at least a 3.5-year follow up [19]. However, the relevance of metformin to counteract drug resistant cancer development has not been well studied. In a recent review, the ability of metformin to inhibit the activity of cancer stem cells was considered [20]. It has been found that metformin can increase the sensitivity of cancer stem cells to chemotherapy through a variety of methods, including improving vascular normalization and suppressing tumor angiogenesis in metastatic breast cancer cells. Thus, it appears that metformin has potential against MDR cancers, but much remains to be learned regarding the molecular mechanisms impacted by metformin under MDR conditions.

The aim of this study was to determine whether metformin can reverse drug resistance observed in lymphoma in vivo as it does in vitro in breast cancer [10]. Canine lymphoma is a proliferation of neoplastic lymphocytes typically causing multicentric lymphadenopathy with additional infiltration of neoplastic cells into other organs, such as liver, spleen, and bone marrow, among other body sites. Most lymphomas can be immunophenotyped to determine either T or B lymphocyte origin. We used an in vivo companion canine model of spontaneous MDR B-cell lymphoma and delivered oral metformin treatment. This disease is orthologous to non-Hodgkin’s lymphoma and has a relapsing-remitting course with a high incidence of terminal MDR transformation [21,22,23]. The superficial nature of lymph node enlargements allows for easy and repetitive non-invasive tumor sampling. Standard clinical management of lymphoma in companion canines involves repeated weekly appointments, providing an opportunity to sample a given tumor over time during the chemotherapeutic regimen. RNA and protein samples were obtained from canine tumors and control tissue both prior to and following metformin treatment, and analyzed via qRT-PCR, microarray, and Western analyses. Matched sets of sensitive and resistant OSW canine lymphoma cells were used for validation experiments. We identified a novel mechanism whereby the Anaphase Promoting Complex (APC), a large multi-subunit ubiquitin ligase that is structurally and functionally conserved from yeast to humans, is required to protect cells from the development of MDR.

## 2. Materials and Methods

### 2.1. Companion Canine Recruitment and Characteristics

Canines were recruited from the Oncology Department at the Western College of Veterinary Medicine between July 2012–July 2015. Dogs with cytologically or histologically confirmed multicentric lymphoma that failed to achieve complete remission with a standard chemotherapy protocol were included in the study upon spontaneous relapse and positive MDR biomarkers (MDR-1). Additional inclusion criteria included the need for owner consent; their ability to financially contribute to the treatment cost, time and travel required to receive chemotherapy; owner willingness to administer oral metformin tablets up to twice daily; free of significant underlying medical disease and a VCOG performance status of 0 or 1; and a life-expectancy of at least 6 weeks. The study protocol was approved after a full review by the University Animal Care Committee-Animal Research Ethics Board at the University of Saskatchewan (AREB# 20120063). We confirm that all experiments were performed in accordance with relevant guidelines and regulations. Pre-treatment evaluations included physical examination; caliper measurements of all accessible lymph nodes; bloodwork (complete blood cell count, serum biochemistry profile) and urinalysis; thoracic radiographs; abdominal ultrasound with fine needle aspirates of liver and spleen; lymph node extirpation for histological examination, immunohistochemistry and protein extraction; biopsy of normal skin for tissue comparison; and aspirate of peripheral lymph node for RNA extraction. Lymph node extirpation and skin punch biopsy were performed under general anesthesia. Tumor samples, obtained by three fine-needle aspiration biopsies taken from a palpable superficial lymph node (the same node could be sampled over time if still palpable), section of lymph node biopsy, and a punch biopsy of unaffected skin, were stored at −80 °C for later mRNA and protein analyses. One FNA sample was placed immediately into Trizol for subsequent RT-PCR and microarray analysis, with the remaining two FNA samples pooled and immediately frozen at −80 °C for Western analysis. Oral metformin tablets were dosed on body weight with a maximum dose of 10 mg/kg, initiated as once-daily and increased as tolerated to twice daily at weekly clinic appointments. Metformin as adjunct therapy is nontoxic and well-tolerated [24,25].

### 2.2. Microarray Hybridization

Total RNA from tumor and skin samples was shipped on dry ice to the Laboratory for Advanced Genome Analysis at the Vancouver Prostate Centre for microarray analysis (http://www.mafpc.ca/ accessed on 25 January 2016). Total RNA was used as a template to create labeled cDNA using MessageAmp Premier RNA Amplification Kit and MessageAmp III RNA Amplification Kit (Applied Biosystems, Waltham, MA, USA) according to the manufacturer’s instructions. Labeled cDNA was hybridized to Agilent Canine Microarrays, which are comprised of more than 25,000 annotated genes. Scanning and data acquisition were obtained using the Illumina iScan scanner, with raw data (idat files) loaded into Illumina BeadStudio, without background subtraction, and exported for analysis. The data files were deposited with the meta-database Gene Expression Omnibus (https://www.ncbi.nlm.nih.gov/geo/query/acc.cgi?acc=GSE121242; GEO accession # GSE121242 accessed on 1 December 2018). Sufficiently high quality of extracted RNA was not available for all samples or at all time points, limiting the overall assessment possible. We were only able to extract RNA from the skin control from Canine 2. Canine 4 was unique in providing multiple quality protein and RNA samples over time as the canine progressed through clinical relapse and regression.

### 2.3. Microarray Data Analysis

The microarray data were analyzed using the Limma R package (https://bioconductor.org/packages/release/bioc/html/limma.html (accessed on 2 August 2022)). Raw probe intensity values were background-corrected using the backgroundCorrect function with method = “normexp” and quantile-normalized using the normalizeBetweenArrays function. X represented the 95th percentile of the intensities of the negative control spots for a given array. We ignored any non-control probe for which there was no array having an intensity value for that probe that was more than 10% higher than X. The expression of a given gene was quantified by averaging the intensity values of all the probes corresponding to that gene. Finally, log2 fold-change values were calculated for each gene and for each treatment-control combination of interest. Venn diagrams were used to identify a common set of 290 genes that were expressed greater than 3-fold in all four canines, compared to skin controls in all four canines. Following identification of the 290 overexpressed genes that were shared between the four MDR canines, gene set enrichment analyses were conducted using Cytoscape 3.7.0 with ReactomeFIViz (2017 ReactomeFI Network Version 62, reactome.org (accessed on 2 August 2022). The 290 gene network was clustered to identify associated genes, and subsequent pathway enrichment was conducted. Pathways with a *p*-value and false discovery rate (FDR) of *p* < 0.05 were included in results. The 290 gene list was also analyzed by STRING, version 10.5 (elixir-europe.org (accessed on 2 August 2022)) as an independent method to identify clustered and associated genes. Analysis of over-represented transcription factor binding sites from the 290 up-regulated gene set was conducted using the Cis-element Over-representation algorithm (CLOVER; downloaded 1 December 2018, https://github.com/mcfrith/clover) with the TRANSFAC transcription factor motif database (2019). Analyses were run using a background sequence comprised of 1500 base pair sequences up-stream of transcription start sites in the genome. Sequences were retrieved from e!Ensembl database, Dog genes CanFam3.1. 1500 base pairs upstream of transcription start sites of each of the 290 genes were examined for the presence of transcription factor binding motifs, and motifs with a *p*-value < 0.05 (overrepresented) and *p*-value > 0.95 (underrepresented) identified.

### 2.4. Cell Lines, Drug Selection, Methods and Materials

Canine OSW lymphoma cells were purchased from American Type Culture Collection (ATCC, Manassas, VA, USA). Cells were cultured in 100 mm tissue culture dishes in 5% CO_2_ at 37 °C, containing RPMI 1640 media (Hyclone, Logan, UT, USA) with 10% FBS and antibiotics. All treatment compounds were reconstituted in dimethylsulfoxide (DMSO, ThermoFisher, Waltham, MA, USA) except metformin (Sigma-Aldrich Canada, Oakville, ON, Canada) which was reconstituted in molecular-grade water and filter sterilized prior to use. Drug treatments were applied at the concentrations and times as indicated. Flow cytometry was performed as described previously [7]. OSW parental cells were selected for drug resistant populations against Doxorubicin (DOX) using established protocols [9]. Protein lysates from cultured cells were prepared according to our previously published methods [7,9]. Lysate preparation from whole tumors was similar, but with longer sonication times. Primary antibodies against MDR-1 (Abcam, 1:500, 180 kDa, Waltham, MA, USA), BCRP (Abcam, 1:1000, ~70 kDa), HIF1α (Santa Cruz Biotechnology (SCBT), 1:500, 93 kDa, Dallas, TX, USA), HURP (Abcam, 1:1000, 95 kDa), Cyclin B1 (Sigma, 1:1000, 70 kD), Securin (Abcam, 1:1000, 22 kDa), GAPDH (Abcam, 1:1000, 55 kDa), and tubulin (Sigma, 1:1000, ~50 kDa), were used in this study. The blots were then decorated with a 1:10,000 dilution of a horseradish peroxidase (HRP) secondary antibody. Membranes for all Westerns were cut to maximize the amount of information for each membrane, therefore, full blots are generally not available. Cell proliferation was measured using MTT (3-(4,5-dimethylthiazol-2-yl) 2, diphenyl-tetrazolium bromide) according to our published protocols [10].

## 3. Results

### 3.1. Companion Canines with Non-Hodgkin-like Lymphoma Are Strong Models of MDR Malignancy

All canines in this study presented to the oncology clinic at the WCVM with spontaneous multicentric lymphoma. Signalment, stage of disease, and immunophenotype results are presented in Table 1. Prior to enrollment in the study, all canines were initially treated with a CHOP-based chemotherapy protocol and achieved a complete clinical remission (all lymph nodes returned to normal size) with variable remission times. The canine was determined to be out of clinical remission when generalized lymphadenomegaly had recurred and resistance to chemotherapy was identified molecularly by MDR gene analysis via the aforementioned methods. Canines that fulfilled these criteria were enrolled in the study. Initial samples taken at the time of enrollment were from peripheral lymph nodes (mandibular, superficial cervical, or popliteal) via fine needle aspirates to determine MDR status. The canine was then started on a CHOP protocol again, with the addition of metformin. We continued to sample lymph nodes (weekly) if the lymph nodes were still enlarged.

Access to these canines allowed us to test our in vitro observations that metformin reversed MDR protein markers and resensitizes MDR cells to chemotherapy [10]. We obtained tumor samples from eight canines in total; two from naïve drug sensitive canines and six that had failed their drug therapy. Details of the four canine subjects used for our complete analyses are shown in Table 1. Canine subjects 1–4 were clinically unresponsive to chemotherapy, and all received adjunct oral metformin, as described. Canine subjects 5 and 6 were treatment resistant, but were not treated with metformin and were not followed any further due to owner decision not to proceed with the clinical trial. Canine subjects 7 and 8 were treatment sensitive, were not staged, and did not receive metformin. No canine achieved a clinical remission with the addition of metformin, however, partial responses were recorded and clinical benefit was reported by owners.

### 3.2. Canines with MDR Lymphoma Overexpress Proteins Associated with Drug Resistance, and Adjunct Metformin Therapy Reverses This

Western analyses were used to determine the relative abundance of the MDR biomarker, MDR-1, in cancerous lymph nodes as compared to skin control samples, before and after metformin use. MDR-1 is a non-specific drug efflux pump from the ABC transporter family (ABCB1). We used fine needle aspirates (FNAs) to obtain tumor samples from five MDR canines (samples 1–4, 6), who all expressed high levels of MDR-1 in their tumor sample (Figure 1, The uncropped Western blots are shown in Appendix A). Canines 7 and 8, as expected, did not express MDR-1 since they were clinically treatment-sensitive (Figure 1B). The MDR protein markers BCRP (Breast Cancer Resistance Protein) and HIF1α (Hypoxia Inducible Factor 1 alpha) were also elevated in Canine 3 (Figure 1C). Samples were then taken at the indicated weeks after metformin initiation with all MDR protein markers showing a rapid decline (Figure 1). The in vitro observations showing loss of MDR markers following metformin treatment [10] are therefore also observed in vivo, and potentially clinically relevant as one canine experienced a period of remission following metformin exposure.

### 3.3. A 290 Gene Set Overexpressed > Three-Fold in Tumor Versus Control Was Common to the Four MDR Canines and Contained 20 Anaphase Promoting Complex Substrates

We isolated RNA from tumor cells and skin biopsies obtained from four MDR canines in our study (Canines 1–4). We performed microarray analyses (Agilent Canine Microarrays; 25,000 annotated genes) to identify genes that were differentially expressed in the tumor samples compared to normal skin cells (Canine 2), and in the tumors before and after metformin treatment (Canines 2 and 4). The datasets were deposited at the GEO repository (GEO accession # GSE121242; https://www.ncbi.nlm.nih.gov/geo/query/acc.cgi?acc=GSE121242 accessed on 1 December 2018). We compared our datasets to genes recently identified as differentially expressed in canines with diffuse large B-cell lymphoma [26]. We found that many of the genes previously identified were similarly elevated in our canines, particularly those in the B-cell receptor, NF-kB and MYC signaling pathways. Next, since the ABC transporters MDR-1 and BCRP were observed elevated at the protein level in the canine MDR tumors (used as biomarkers of MDR), we analyzed differential expression of the remaining 40 ABC transporters on the array, and only *ABCB3* was elevated above three-fold change (FC) in all MDR canines at the mRNA level (Appendix A). *ABCB1* mRNA (encoding MDR-1) was above 3 FC in only one tumor sample (Canine 4), suggesting that increased MDR-1 protein levels may occur post-translationally. Next, we averaged expression from the tumor samples before and after metformin treatment from canines 2 and 4 (Appendix A). Consistent with metformin reducing the protein levels of MDR-1 (Figure 1), expression of *ABCB1* mRNA was also reduced by metformin in the two canines. Four other ABC transporters showed this pattern, with elevation in the MDR tumor at least two-fold greater than in the control sample, and reduced by metformin (*ABCA3*, *ABCC10*, *ABCG1*, and *ABCB3*). We concluded that it was unlikely that the ABC transporters were playing a role in MDR development and looked for other common differentially expressed genes in the tumors of the four MDR canines (Canines 1–4).

Between 875 (Canine 2) and 1365 (Canine 3) genes were overexpressed >three-fold in the four canine datasets as compared to noncancerous control tissue. We identified a common set of 290 genes overexpressed greater than 3 FC in all four canine datasets (Figure 2A; Appendix A). Using Cytoscape and Reactome FI [27], we discovered that 146 of these genes were highly associated. In total, 13 different gene clusters were identified (as defined by different color spheres and asterisks; Figure 2B) and are listed in Appendix A. Using the software ‘Search Tool for the Retrieval of Interacting Genes/Proteins’ (STRING, version 10.5), we identified 186 genes from the 290 gene set that were highly interconnected (Appendix A). A particularly tight interconnected gene cluster (shown in forest green (i.e., DLGAP1) in Figure 2B; yellow nodes in Appendix A, highlighted in yellow in Appendix A) was identified, which was involved in multiple aspects of mitotic regulation. Other clusters also contained genes involved in mitotic progression, DNA repair, and DNA replication (green and purple nodes in Appendix A; highlighted in green in Appendix A). These observations demonstrate that overexpression of a tightly connected network of genes involved in cell cycle progression and genome integrity may drive MDR development.

Both Cytoscape with ReactomeFIViz and STRING identified the same cluster of mitotic genes, in which many of the genes encoded protein degradation substrates of the Anaphase Promoting Complex (APC; 20 genes, marked by *, Appendix A; nodes circled in red in Figure 2B). The APC is a conserved ubiquitin E3 protein ligase that targets proteins that inhibit cell cycle progression for ubiquitin- and proteasome-dependent degradation [28,29]. APC substrate mRNAs often peak during G2/M, then decline once the protein is degraded [30,31]. Several APC substrates are transcription factors that themselves drive the transcription of other APC substrates [32,33,34]. Thus, elevation of these APC substrates upon APC impairment will maintain the high levels of other substrate mRNAs. Furthermore, proteins associated with the Spindle Assembly Checkpoint (SAC) and the kinetochore, which predominantly inhibit APC activity prior to mitosis [35], were elevated (15 genes, marked by **, Appendix A). Moreover, proteins that maintain chromosome condensation (the condensin complex) were also elevated (three genes; marked by ***, Appendix A). These observations suggest that delayed anaphase progression mediated through APC inactivation may be key for the onset of MDR.

### 3.4. Mitotic and G1 APC Substrate Gene Expression Is Elevated in Treatment Resistant Tumors Compared to Normal Controls

To test this idea, we began by separating the known APC targets on the array into M vs. G1 targets (11 mitotic substrates and 21 G1 substrates), based on PubMed searches (see [36] for full analyses), to assess if there were differences in expression between M and G1 substrates (Figure 3A). Different coactivators interact with the APC to promote cell cycle progression; CDC20 during mitosis (APC^CDC20^); and CDH1/FZR1 for mitotic exit and transit through G1 (referred to as APC^CDH1^ hereafter) [28,29]. CDC20, an APC activator and substrate, is often overexpressed in cancer cells [37,38], while impairment of CDH1 leads to genome instability and cancer development [39,40,41]. Furthermore, mutation to APC subunits is associated with resistance against some chemotherapeutic agents [42]. We consistently observed that the mitotic targets were overexpressed in MDR tissue to a greater extent on average than that observed for G1 substrates within all four MDR samples (Figure 3B).

Coordinated APC substrate gene expression may be due to activation by one or more common transcription factors. To test this possibility, we implemented a Cis-element Over-representation (CLOVER) Analysis [43] using the TRANSFAC database to analyze the 290 gene list. This analysis will identify over-represented transcription factor binding sites within gene promoters of our overexpressed genes of interest. We observed that 230/290 of the genes have FOXO3A and/or FOXM1 binding sites, and 241/290 genes have sites for the SP1-4 transcription factor family. A Venn diagram shows that 225 of the genes contain sites for all three sets of transcription factors (Figure 3C). Our finding that FOXM1 may be highly involved in the expression of the 290 gene list is significant since FOXM1 is a known APC^CDH1^ substrate [44] and FOXM1 expression was elevated in all canine tumors (Figure 3A).

### 3.5. Microarray Reveals Reversible Changes in APC Target Gene Expression That Correlate with Altered Clinical Treatment Responses

Canine 4 had failed both CHOP and rescue therapy in the months prior to our study entrance, but exhibited a successful partial remission based on the near disappearance of all palpable lymph nodes following adjunct metformin therapy combined with a repeat CHOP protocol (Table 2). After 10 weeks in remission, the subject relapsed and expired. RNA samples were obtained from the same lymph node tumor location prior to metformin therapy, during remission, and after relapse. From these RNAs, we identified a set of 27 genes that were upregulated >3 FC in the MDR tumor, decreased >2 FC upon remission, and then elevated once again >3 FC upon relapse compared to the skin control (Figure 4A). We performed STRING on this set of genes and found that they were highly enriched in APC substrates that were overexpressed at a time of clinical unresponsiveness to therapy (Figure 4B, highlighted with a black ring around the node; Appendix A). Further analyses of APC mitotic and G1 substrates in our three temporal samples from Canine 4 revealed a similar pattern (Figure 4C). We validated the gene expression changes using qRT-PCR on the original FNA samples from chemo-resistant and remission samples in Canine 4, for the genes encoding HURP (*DLGAP5*) and Securin (*PTTG1*; Figure 4D). This provided support for the concept that APC function, affecting its substrate abundances, is correlated with clinical disease responsiveness.

### 3.6. APC Substrates Are Elevated In Vitro in OSW Lymphoma Cells Selected for DOX Resistance and Reversed When the APC Is Activated

Next, we obtained the canine OSW lymphoma cell line [45] to test whether our in vivo observations were valid in vitro. We selected OSW cells for resistance to DOX (OSW^DOX^) according to our established methods [9,10]. The MDR status of the OSW^DOX^ cells was confirmed by the overexpression of several protein biomarkers of MDR (BCRP, MDR-1, and HIF1α; Figure 5A), and the detection of significant resistance to DOX cytotoxicity (Figure 5B). Quantitative PCR (qPCR) comparing OSW parental (OSW^sens^) and OSW^DOX^ confirmed that APC substrate genes *PTTG1*, *DLGAP5*, and the MDR marker *MDR1* were elevated at the mRNA level in MDR cells in vitro (Figure 5C). As expected, with a defect in APC activity, we observed that APC protein substrates (HURP, Cyclin B1, and Securin) were increased specifically in the MDR cell populations (Figure 5D,E).

If impaired APC activity correlates with MDR development, we predicted that activating the APC may reverse the MDR phenotype by resensitizing these cells to chemotherapy. To directly investigate this idea, we obtained a commercially available APC activator called MAD2 Inhibitor-1 (M2I-1) [46]. This small molecule compound disrupts the MAD2-CDC20 interaction, resulting in the premature release of CDC20 from the SAC to enable its interaction with, and activation of, the APC. We treated OSW^DOX^ cells with 1 μM M2I-1 for 18 hours and assessed the protein levels of the APC substrates HURP and Securin, and the load control tubulin (Figure 5F). HURP and Securin protein abundance decreased upon M2I-1 treatment in OSW^DOX^ cells, to levels lower than in OSW^sens^ cells (Figure 5G), indicating that APC E3 activity was being restored.

To measure the impact of M2I-1 on cancer cell chemosensitivity, we treated OSW^sens^ and OSW^DOX^ cells with DOX (1 μM) and/or M2I-1 (1 and 5 μM). M2I-1 was nontoxic to chemosensitive and MDR cell populations alone, however, there was a striking dose-dependent killing of OSW^DOX^ cells when M2I-1 and DOX were used in combination that reached the chemosensitivity of OSW^sens^ cells (Figure 5B). A potentially compromised APC, such as in MDR cell populations (i.e., OSW^DOX^) may result in M-phase accumulation with a 2n compliment of DNA since the APC cannot properly drive mitotic exit. However, this was clearly not the case, as flow cytometry showed that both OSW^sens^ and OSW^DOX^ cells were predominantly in G1 (Figure 5H). Thus, the enrichment of mitotic cell cycle genes in MDR cells appears to be due to the chemo-resistant state of the cells, and not an increased mitotic index. Taken together, our results indicate that APC activity is impaired in MDR cells, and that chemical activation of the APC resensitizes them to chemotherapy.

## 4. Discussion

Our goal was to determine if MDR could be reversed in vivo, similar to our observations in vitro using the insulin sensitizer, metformin [10]. In addition, we sought to elucidate the molecular mechanisms involved in the development of MDR and its reversal. Our previous in vitro work demonstrated that the insulin sensitizing medication metformin could restore the chemosensitivity of human MDR breast cancer [10]. Furthermore, pretreatment of MCF7 cells with metformin strongly inhibited the development of MDR despite strong selection pressure. This is consistent with meta-analyses indicating that individuals taking metformin for type 2 diabetes not only have a lower incidence rate of many common cancer types, but also appear to respond more robustly to therapy [47,48]. Here, we recruited six canines presenting to the WCVM with spontaneous MDR lymphoma; all four MDR canines treated with metformin had reduced protein markers of MDR (Figure 1) and one of the canines went into a partial remission after failing all other rescue protocols. Microarray analyses from tumors from four of these animals revealed that APC-substrate mRNAs were elevated (Figure 2A and Figure 3A,B), consistent with impairment of APC activity. In fact, promoter binding sites for the APC substrate FOXM1, found elevated in MDR tumors, were enriched in the common 290 gene set elevated in the four canines (Figure 3C). In vitro validation of the in vivo results used matched sensitive and resistant OSW canine lymphoma cancer cells, which confirmed our results: (i) resistant cells had higher mRNA and protein levels of APC substrates, (ii) activation of the APC using a small indirect chemical APC activator (M2I-1 [46]) reduced APC substrate protein levels (Figure 5F,G), and (iii) APC activation resensitized cells to chemotherapy (Figure 5B). Translation of our in vitro results to a clinical model, whereby targeted APC activation is trialed as a therapy, could provide enormous potential for humans who develop resistance to their therapies.

The canine model of cancer is considered an excellent model for human translation [49,50,51]. The benefits include their genetic diversity, their large body size, an intact immune system, and spontaneous tumors with similar biology and tumor burden to humans. Furthermore, we could repeatedly sample the same tumor over time non-invasively and monitor clinical responses. The MDR canines in this study overexpressed a variety of MDR protein biomarkers (such as MDR-1, an ABC drug efflux transporter found elevated in many cases of MDR cancers in humans [9,52]) and treatment with metformin reduced MDR protein levels, analogous to our in vitro observations [10]. One canine (Canine 4), after failing multiple treatment rounds and all previous rescue therapies, achieved a strong partial remission (Table 2). However, it should be noted that the canines enrolled in our study were terminally ill, and although there were signs of improvement in their behavior, there was little sign of reduced tumor size aside from Canine 4. Future studies will focus on naïve dogs as they enter the clinic for a blinded study with metformin to determine whether early introduction of metformin can delay the onset of drug resistant lymphoma.

We obtained useable tumor RNA from four dogs with matched, non-malignant skin sample prior to metformin use, which was applied to canine microarrays. Repeat RNA samples were collected for several weeks after metformin was initiated (Canines 2 and 4), as well as repeatedly from the one canine (Canine 4) that entered remission and later relapsed. Our analyses of overexpressed genes in the MDR tumors present in all MDR samples (without metformin) revealed a common set of 290 genes overexpressed at least three-fold (Figure 2A). Use of Cytoscape and STRING databases to analyze this gene set revealed 146 and 186 of the 290 genes, respectively, were highly associated (Appendix A). We focused on highly enriched genes encoding APC substrates, but also note the enrichment of genes encoding components of the kinetochore and SAC that inhibit the APC. Other genes that encode proteins required for DNA-dependent functions (i.e., DNA repair, chromosome condensation and kinetochore/centromere assembly) were also enriched, indicating a shift towards enhanced chromosome maintenance in MDR cells. These gene clusters highlight that MDR development is, unsurprisingly, multifactorial but may unexpectedly rely on slowed progression through mitosis (facilitated through impaired APC function) to enable DNA repair and chromosome segregation in cells despite a high load of DNA damage and chromosomal instability [53,54].

Supporting the idea that the APC is impaired specifically in treatment resistant, rather than treatment sensitive cases, we found that many of the genes elevated at least 3 FC on the array were elevated in all four canines (Figure 3). Previous analyses in a variety of human cancer cell lines had also demonstrated elevated mRNA expression encoding for numerous APC substrates [55,56]. Furthermore, a recent comparison of drug sensitive and resistant ovarian tumor datasets [57] revealed a collection of genes with differential expressions similar to ours; half of the 30 highly connected nodes identified were associated with APC function or substrates. Consistent with these observations, remission and relapse of Canine 4 following metformin treatment identified 27 genes that were upregulated in the tumor, down-regulated following metformin treatment, then upregulated again following relapse (Figure 4A). These 27 genes were highly interconnected and predominantly composed of APC substrates (Figure 4B), placing the APC at a potential pivot point as a critical activity in maintaining cell health.

Elevated APC protein substrates can arise from multiple mechanisms, including failure to be degraded due to dysfunctional APC E3 ligase activity, but also due to dysregulation of their gene expression. The APC targets proteins for degradation, not mRNA (as identified in our arrays), but APC substrates are often transcriptionally active just prior to when they are required [30,31]. This is controlled by transcription factors that are themselves degraded during the cell cycle by the APC, including FOXM1, an oncogene that is a marker of poor patient prognosis [32], which was found to have promoter recognition sites in many known APC target genes [44], and in 230 out of 290 gene set (Figure 3C). The mRNAs encoding many APC substrates in yeast, such as the transcription factors Fkh1 and Ndd1 [33,34], are also cell cycle regulated, with synthesis of their corresponding proteins peaking following the mRNA expression peak [58,59,60]. The accumulation of APC substrate mRNA, at least in part, is a result of impaired APC-dependent degradation of the transcription factors responsible for their synthesis.

A failure to appropriately degrade APC protein substrates in synchrony with the cell cycle can have many biological consequences that are relevant to cancer behavior. First, many substrates, such as CCNB2, PLK1, and CDC20, are both APC activators and APC substrates. Elevation of these mRNAs and proteins could result in persistent APC activity which impacts the cell cycle and proliferation. Increased accumulation of these APC substrates in many aggressive cancers has led to suggestions that therapies that down-regulate these targets, such as CDC20 or others, may be important for augmenting cancer treatment responses [37,38]. However, it is important to note that APC activity is essential [61,62], therefore, systemic impairment of the APC could have catastrophic effects on the organism. On the other hand, classical downstream substrates, such as Securin, HURP, and many others, are independently linked with cancer and other diseases [54,55,56], which is consistent with the idea that impaired APC activity is a primary driver of tumor progression [39,40,41,42]. Using the Cancer Genome Atlas (TCGA) database (https://portal.gdc.cancer.gov/ accessed on 27 April 2018) we show that the APC substrates Cyclin B1 (Appendix A) Securin, and HURP [54] are elevated in at least 24 different human cancers. The numerous APC substrates found to be elevated in this study suggests a global defect in APC activity, which would be predicted to slow mitotic progression, allowing time to repair sufficient DNA damage in cells with high loads of chromosome instability (CIN) [62,63]. Slowed progression through anaphase in MDR cells is supported by high expression of genes encoding SAC components and the kinetochore, which inhibit the APC prior to anaphase [35,64,65] (Appendix A). Elevated expression of SAC and kinetochore components are also associated with aggressive cancer [36,66,67,68,69], with SAC inhibitors now entering Phase I clinical trials [70,71].

In vitro validation using canine OSW^DOX^ lymphoma cells confirmed that APC substrates were elevated in MDR cells compared to parental chemosensitive cells (Figure 5B–D). More importantly, activation of the APC using a commercially available small compound that inhibits the MAD2-CDC20 interaction, M2I-1 [46], reduced APC substrate levels in DOX selected cells and resensitized them to DOX (Figure 5B,F,G). Interestingly, we observed that OSW^DOX^ cells are primarily in G1 with elevated levels of APC mitotic substrates (Figure 5H). A previous study of 182 breast cancer patient samples revealed that 58% of cases characterized contained elevated levels of the mitotic APC substrates, yet stained abundantly for G1/S markers [72]. These samples were from patients with high grade triple negative breast cancers and at high risk of relapse. Cells with elevated mitotic markers but in G1 can occur by mitotic slippage, where cells can bypass a mitotic arrest and progress into G1 [73]. Mitotic slippage can occur when the SAC is overridden, and this is associated with an MDR phenotype. Our work revealing that OSW^DOX^ cells exhibit high levels of APC mitotic substrates in a population predominantly in G1 is consistent with a higher grade cancer and has the potential to serve as a diagnostic marker for aggressive tumor progression.

## 5. Conclusions

Taken together, the results presented here demonstrate that the companion canine is a powerful and dynamic model of MDR cancer. Using this model, and validation using in vitro cell lines, we observed that impaired APC activity correlates with poor clinical outcomes, as well as MDR behavior. Exogenous activation of the APC corrected the impairment and restored chemosensitivity in vitro. Our work provides insight into the APC as a novel therapeutic target that may offer hope for individuals presenting with treatment-resistant malignancies and may prove useful in preventing the development of resistance in cancer patients in the future.

## Figures and Tables

**Figure 1 cancers-14-04215-f001:**
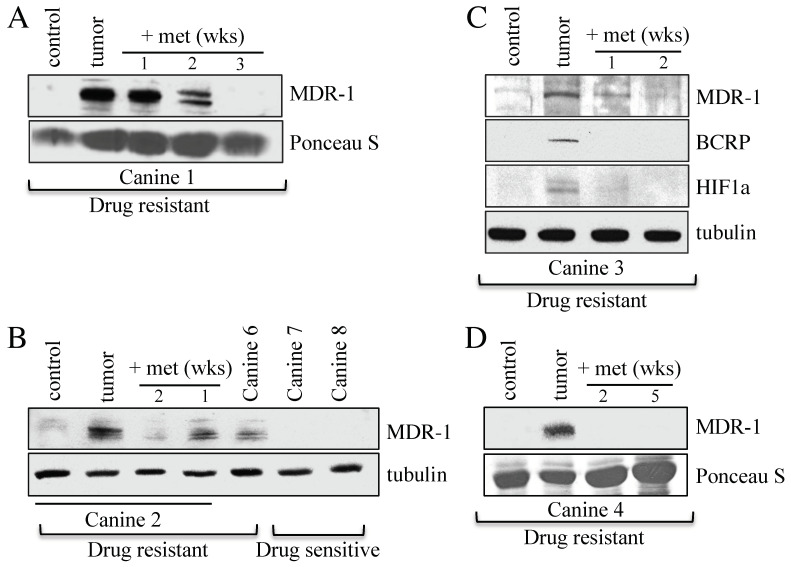
Changes in MDR protein biomarker levels within tumor samples from canine subjects before and after oral metformin addition. (**A**) Protein lysates were prepared from Canine 1 that presented with MDR lymphoma. The lysates were analyzed by Western blotting using antibodies against MDR-1. Ponceau S staining was used to ensure equal protein load. The control was derived from skin samples from Canine 2. (**B**) Protein lysates were prepared from Canines 2, 6, 7, and 8 for analysis by Western blotting using antibodies against MDR-1 and tubulin. (**C**) Lysates prepared from tumor samples obtained from drug resistant Canine 3, before and after metformin therapy, were analyzed with antibodies against the MDR markers shown, with tubulin serving as the load control. (**D**) Protein lysates prepared from tumor samples from drug resistant Canine 4, before and after metformin treatment, were assessed using antibodies against MDR-1, with Ponceau S staining representing relative load controls.

**Figure 2 cancers-14-04215-f002:**
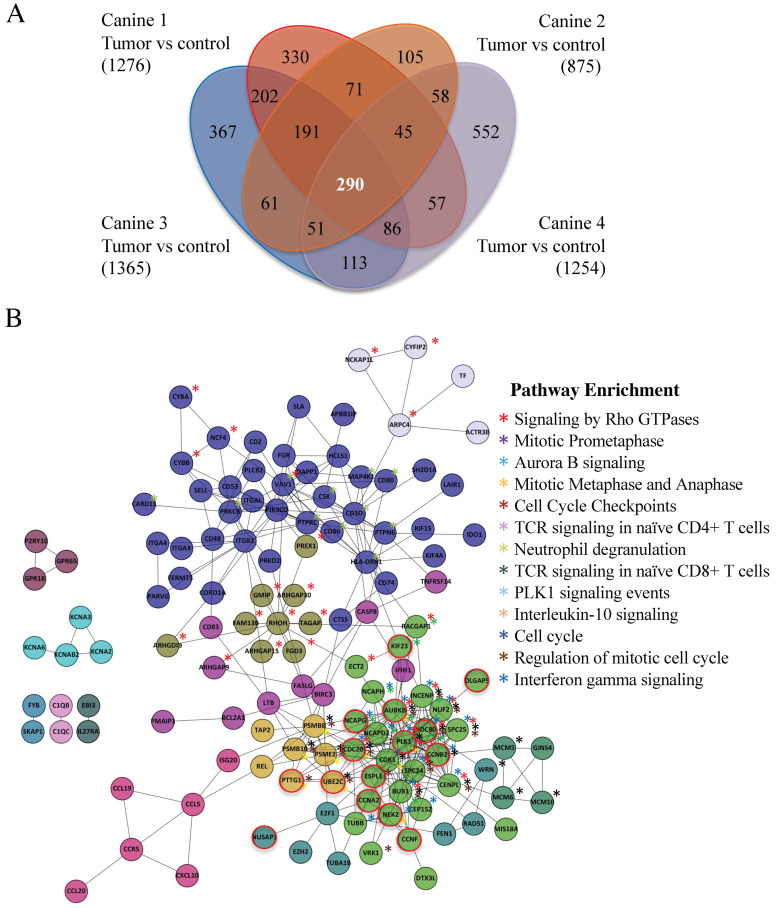
Canines with MDR lymphoma tumors express a common set of highly interactive genes. (**A**) A Venn diagram highlights a set of 290 genes overexpressed in the tumors of the four MDR canines studied. Agilent canine arrays containing over ~25,000 annotated canine genes were used to analyze mRNA obtained from tumor samples in four MDR subjects (Canines 1–4), which were compared to control skin tissue mRNAs (Canine 2). The fold change (FC) was determined by comparing the Log base 2 expression levels from the tumors with the control. A Log base 2 of 1 is equivalent to a FC of 2. A FC > 3 was used as a cut-off. Bracketed numbers reflect the total number of genes above the FC 3 threshold in that subject. (**B**) Cytoscape was used to analyze the 290 gene set. Further, 146 genes were found to form 13 highly interconnected nodes. Network pathways found to be enriched within these nodes are shown. Astericks denoting genes within the nodes that are enriched within network pathways are color coded. Nodes circled in red define Anaphase Promoting Complex (APC) substrates.

**Figure 3 cancers-14-04215-f003:**
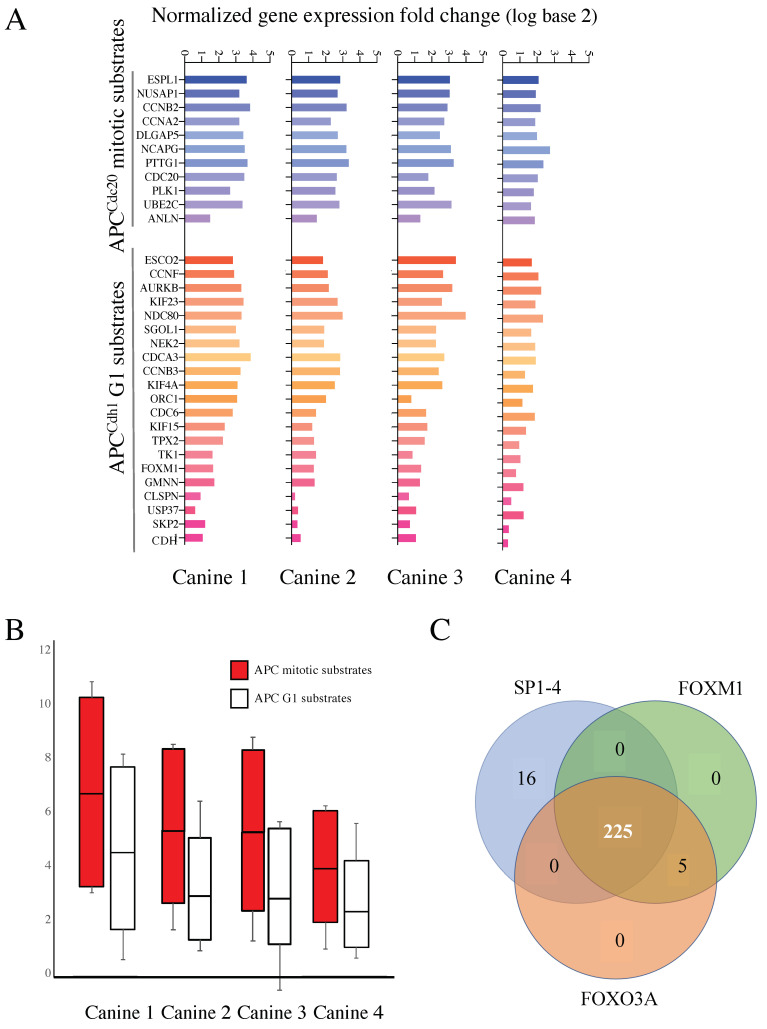
mRNAs encoding APC substrates are elevated in the four MDR canines and potentially driven by a common transcription factor. (**A**) All known APC mitotic and G1 specific substrates, to the best of our knowledge, that were present in the canine microarray, were assessed for differential gene expression in MDR tumor samples compared to control skin samples. The histogram in shades of blue reflects APC^Cdc20^ mitotic substrates, while those in shades of red define APC^Cdh1^ targeted G1 substrates. Samples from the four MDR canines are shown individually. (**B**) The scores for all mitotic and G1 substrates from the four canine samples were averaged and plotted as shown. The top and bottom of the box define the third and first quartiles, respectively, while the median of the data divides the box. The whiskers define the error bars for the dataset. (**C**) A Clover analysis using the TRANSFAC database, identified enrichment for FOXM1 (230 genes), FOXO3A (230 genes), and SP1-4 (241 genes) promoter binding sites. A Venn diagram was used to identify overlaps between these datasets.

**Figure 4 cancers-14-04215-f004:**
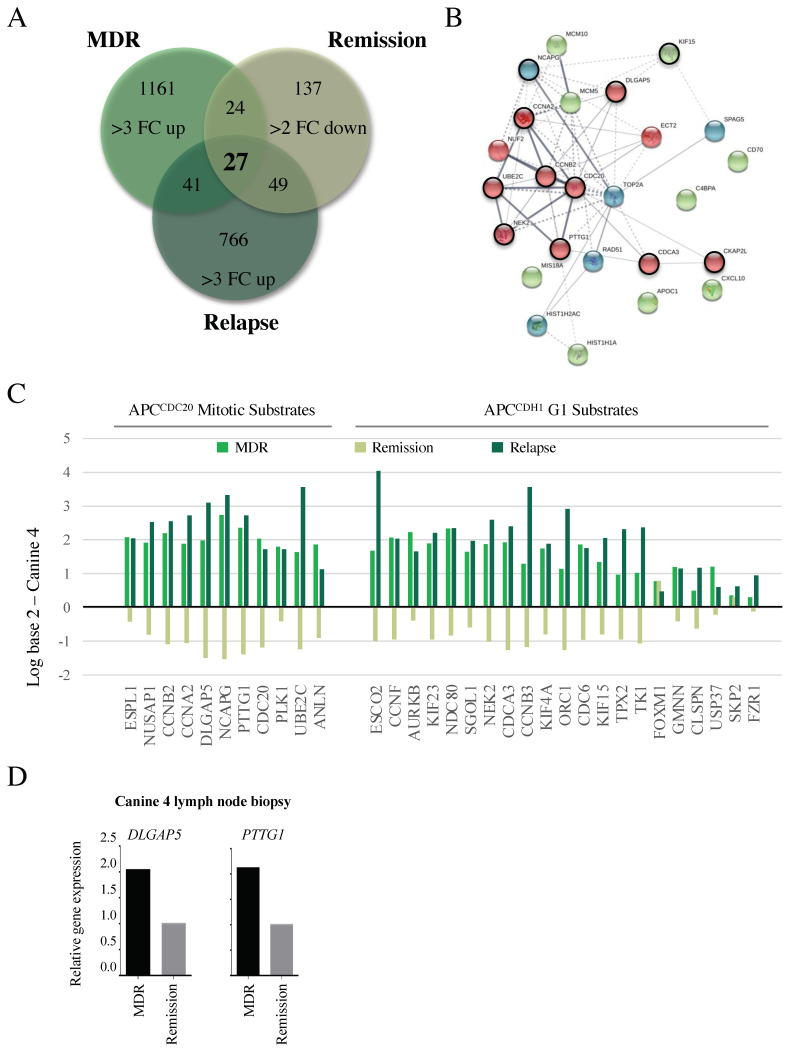
Changes in APC target gene expression correlate with altered clinical responses. (**A**) A Venn diagram was used to determine similarities in genes overexpressed 3 FC in MDR tumors, 2 FC down-regulated during remission, then 3 FC upregulated following relapse. This is predicted to identify genes specifically involved in the MDR phenotype that correlate with clinical responsiveness. (**B**) A STRING analysis indicates that the 27 genes were highly interconnected, with the majority of the genes encoding APC substrates (red nodes, circled in black), and genes encoding proteins required for chromosome maintenance (green and blue nodes, APC substrates circled in black). (**C**) Differential gene expression was determined for the APC substrates present in the array (Figure 3) compared to control skin samples, as correlated with clinical response to treatment at the time of sampling: MDR, remission, and relapse. (**D**) Microarray results were validated by qRT-PCR of original FNA aspirate samples for *DLGAP5* (encoding HURP) and *PTTG1* (encoding Securin) in Canine 4 at study entry and remission.

**Figure 5 cancers-14-04215-f005:**
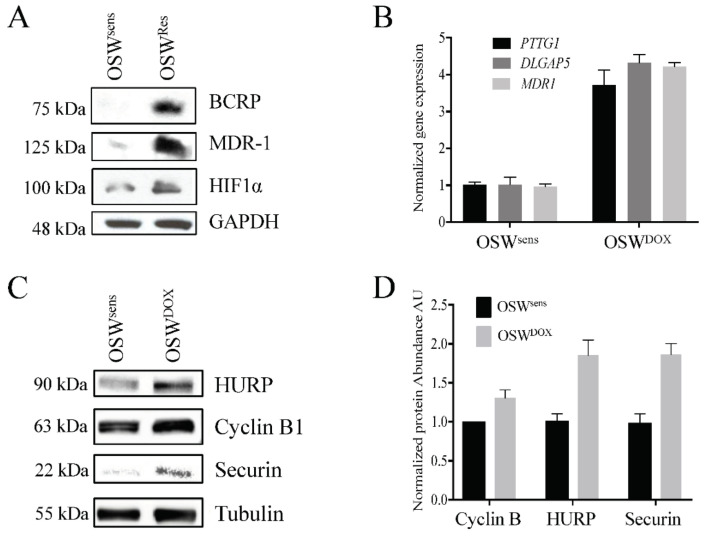
Reversible APC dysfunction in MDR cell populations is validated to occur in vitro and correlates with chemosensitivity. (**A**) OSW canine lymphoma cells (OSW^sens^) were selected for resistance to DOX (OSW^DOX^). Protein lysates from chemosensitive and resistant populations were analyzed by Western analysis for multiple MDR biomarkers. (**B**) OSW^sens^ and OSW^DOX^ cells were pretreated with 1 or 5 μM M2I-1 for 18 hours, then exposed to 1 μM Doxorubicin for 48 hours. Cell viability was measured using Trypan Blue. Three biological replicates were performed. (**C**) qRT-PCR analysis of APC substrate and MDR marker genes in matched OSW populations (*n* = 3). (**D**) OSW cell lysates from (**A**) were tested for APC-target protein abundance. (**E**) Quantification of Western protein abundance using ImageJ, normalized to Tubulin levels (3 rpts). (**F**) OSW canine lymphoma cells selected for DOX (OSW^DOX^) were exposed to the APC activator M2I-1 (1 μM) for 18 hours or left untreated. Untreated parental cells (OSW^sens^) were used as a comparison. Protein lysates from these cells were analyzed by Westerns for the abundance of APC protein targets. (**G**) Protein bands in (**F**) were quantified using ImageJ, normalized to Tubulin levels, and plotted. Three separate Westerns were analyzed. (**H**) Flow cytometry of sensitive and resistant OSW asynchronous cell populations was performed to determine distribution of cell cycle phases. Propidium iodide fluorescence was measured to detect DNA content (*, *n* = 1; **, *n* = 2).

**Table 1 cancers-14-04215-t001:** Clinical characteristics of the MDR lymphoma canines used for the microarray analysis. All 4 MDR canines were male/neutered with recurrent B-cell lymphoma. All canines were clinically nonresponsive/MDR to therapy at the time of enrollment. Skin biopsy samples were taken prior to metformin treatment. Repeat FNA sampling was performed before and during MET addition to chemotherapy. MET: oral metformin in tablet form. OD: once daily. BID: twice daily. Stage III is generalized lymph node involvement of the front and back half of the body. CHOP: 4 drug chemotherapy cocktail given over 19 weeks including Cyclophosphamide, Doxorubicin (Adriamycin), Vincristine, Prednisone. CCNU is lomustine, an alkylating agent.

Case ID Number	Breed	Age	B vs. T Cell Lymphoma	Stage	Chemotherapy at Time of Sampling	Clinical Response to Therapy	MET Duration	MET Dose	Time to Death After MET	Overall Survival (Diagnosis to Death)
1	Golden Retriever	10 yr	B	Ⅲ	CHOP with MET	No, enlarged lymph nodes remained	108 days	250 mg OD	184 days	254 days
2	Lab Retriever mix	5 yr	B	Ⅲ	CHOP with MET	No, enlarged lymph nodes remained	81 days	500 mg BID	87 days	277 days
3	Retriever mix	7 yr	B	Ⅲ	DOX with MET	No, enlarged lymph nodes remained	138 days	500 mg BID	143 days	184 days
4	Lab Retriever mix	11 yr	B	Ⅲ	CHOP failed then rescue with CCNU failed, then MET added to CHOP and remission	Yes/No, Remission attained for 8 weeks then relapse again	84 days	250 mg OD then 250 mg BID	91 days	261 days

**Table 2 cancers-14-04215-t002:** Clinical timeline for Canine 4 treatment. Canine 4 was the only canine to enter remission following the initiation of adjunct metformin. The treatment days are shown along with the treatment received and response to the treatment.

Date	Canine 4 Treatment
29 October 2014	Initial diagnosis by referring veterinarian and initial consult with WCVM oncology.
5 November 2014	Returned to WCVM for staging, started CHOP chemotherapy protocol, received 4 treatments (1 cyce) but never achieved a complete remission.
11 December 2014	Not in remission, abandoned CHOP protocol and started rescue protocol using L-asparaginase and CCNU. Received 5 doses of CCNU along with 2 doses of L-spar. Last treatment was Marth 10, 2015. Achieved a complete remission.
24 April 2015	Out of remission. Fine needle aspirate of lymph node confirmed lymphoma in nodes.
28 April 2015	Enrolled in metformin study, full bloodwork (CBC/biochemistry/urinalysis), abdominal ultrasound, chest X-ray, lymph node removal and skin biopsy.
29 April 2015	Started CHOP again-received first dose of Vincristine and metformin. Went through 2 cycles of CHOP (8 treatments total). Achieved strong partial remission but lymph nodes never completely returned to normal, so there were opportunities to take samples.
13 July 2015	No longer responding to CHOP protocol, progressive disease in lymph nodes, received one dose of CCNU.
27 July 2015	Euthanasia.

## Data Availability

Datasets containing the canine microarray results can be found at the meta-database Gene Expression Omnibus (https://www.ncbi.nlm.nih.gov/geo/query/acc.cgi?acc=GSE121242; GEO accession # GSE121242). Data was deposited 1 December 2018.

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
