# Peer review of "Activation of the Anaphase Promoting Complex Reverses Multiple Drug Resistant Cancer in a Canine Model of Multiple Drug Resistant Lymphoma"

_cancers, 2022, doi:10.3390/cancers14174215_

Round 1

Reviewer 1 Report

This study evaluated a possible strategy for reverting chemo-resistance in tumor cells. This topic is of high impact on both veterinary and human oncology medicine. Dogs could be a perfect animal model of spontaneous malignancies which shared with their human counterparts, treatment, and molecular pathways. 

I recommend the corrections and clarification of some aspects regarding the in-vivo parts of the project. 

Introduction:

Lines 61-67: Those sentences report some material and methods (at which hospital the dogs were included) and discussion (why lymphoma was chosen). Please, rewrite the sentence summarizing information about B-cell multicentric lymphoma in dogs for scientistic working outside veterinary oncology.  

Lines 61-75: the aims of the project are not clearly stated. I suggest to re-write the paragraph and listing the study’s objectives. 

Materials and Methods:

Line 78: Reported here at which hospital the dogs were included. 

Reported if the number of dogs included was chosen for statistical purposes or, in case of no previously decided number, the time lapse for dogs recruitment (E.g., Canines were recruited from the Oncology Department at the Western College of Veterinary Medicine (WCVM) from march 2020 and June 2022)

Clearly list the inclusion criteria of the dogs - How was B cell lymphoma diagnosed: by immune-cytology, -histology, flow cytometry? Which type of B-cell lymphoma were included? How were the dogs staged?

Line 89: what do you mean for renal function testing?

Line 93: considering that the unaffected skin biopsy is outside the common clinical procedure in a dog with lymphoma, please briefly summarize the anesthetic and analgesic protocol applied for this additional procedure. It was not clear if skin biopsy was performed only at admission or, if repeated, at which time point (every week in association with lymph node FNA?) .  

Results: 

Table 1: not found. Please check the manuscript and include table 1. 

If not included in table 1, please report signalment of the dogs. 

Lines 168-169: Report exactly the staging of the dogs included. 

Line 170: The authors reported that all dogs received a first-line CHOP-based protocol before assessing the resistance to chemotherapy, but in Material and methods they stated that dogs received at least 2 different protocols – please, clarify this point.

Lines 171-172: How was the resistance to chemotherapy determined? Please add to the materials and methods section.

Lines 178-184: Were also dogs with tumors sensitive to chemotherapy included? Please verify the correspondence between material and methods and results.  

Lines 182-183: Why were dogs 5 and 6 included in the study if not treated with metformin? 

Table 2: not found. Please check the manuscript and include table 2

Author Response

We thank the reviewers for their careful, timely, and detailed comments on the manuscript. We have incorporated most of their comments. See below for detailed comments when addressing reviews.

Reviewer 1.

Lines 61-67: Those sentences report some material and methods (at which hospital the dogs were included) and discussion (why lymphoma was chosen). Please, rewrite the sentence summarizing information about B-cell multicentric lymphoma in dogs for scientistic working outside veterinary oncology.  

Additional information about canine multicentric lymphoma has been added.  Information on the hospital has been moved to M&M section. Additional information and references regarding metformin activity towards cancer has also been added. See the second and third paragraphs of the introduction.

Lines 61-75: the aims of the project are not clearly stated. I suggest to re-write the paragraph and listing the study’s objectives. 

New text has been added to start the last paragraph of the Introduction.

Materials and Methods:

Line 78: Reported here at which hospital the dogs were included. 

Hospital information has been moved here.

Reported if the number of dogs included was chosen for statistical purposes or, in case of no previously decided number, the time lapse for dogs recruitment (E.g., Canines were recruited from the Oncology Department at the Western College of Veterinary Medicine (WCVM) from march 2020 and June 2022)

There was no previously decided number.  We have included the timeline for enrollment in the study.

Clearly list the inclusion criteria of the dogs - How was B cell lymphoma diagnosed: by immune-cytology, -histology, flow cytometry? Which type of B-cell lymphoma were included? How were the dogs staged?

Further details of inclusion criteria and staging have been added.

Line 89: what do you mean for renal function testing?

This has been changed to serum biochemistry that includes BUN, creatinine, standard electrolyte panel, liver panel and pancreatic markers (lipase, amylase).

Line 93: considering that the unaffected skin biopsy is outside the common clinical procedure in a dog with lymphoma, please briefly summarize the anesthetic and analgesic protocol applied for this additional procedure. It was not clear if skin biopsy was performed only at admission or, if repeated, at which time point (every week in association with lymph node FNA?) .  

The study was approved by the University of Saskatchewan’s Animal Care Committee – Animal Research Ethics Board.  Once the patient was enrolled in the study, full staging was performed including removal of a peripheral lymph node (usually the popliteal if affected) and a skin punch biopsy site separate from the lymph node site.  These procedures were performed by a board-certified veterinary surgeon under general anesthesia.  The patient was given pain control during and after the procedure and sent home with an NSAID.  The authors believe that this information is superfluous for this type of publication and detracts from the readers’ focus.  Similar publications do not have this information explicitly described in the body of the manuscript, and appropriate specimen collection patient management and care are implicit in the approval of the project by an impartial university ethics board. 

Results: 

Table 1: not found. Please check the manuscript and include table 1. 

Table 1 has been added. We apologize for its original omission.

If not included in table 1, please report signalment of the dogs. 

Done. See current line 207 for new text.

Lines 168-169: Report exactly the staging of the dogs included. 

Added.

Line 170: The authors reported that all dogs received a first-line CHOP-based protocol before assessing the resistance to chemotherapy, but in Material and methods they stated that dogs received at least 2 different protocols – please, clarify this point.

All dogs were initially treated with a CHOP based protocol either at the WCVM or at a different clinic.  2 of the 4 dogs did receive a rescue protocol (CCNU and L-asparaginase) prior to enrollment in the study.  The other 2 dogs did not complete the initial CHOP protocol due to coming out of remission.  They were enrolled in the study at that time.  We have removed the sentence stating that dogs received at least 2 different protocols.

Lines 171-172: How was the resistance to chemotherapy determined? Please add to the materials and methods section.

Added and moved to M&M.

Lines 178-184: Were also dogs with tumors sensitive to chemotherapy included? Please verify the correspondence between material and methods and results.  

These canines were not included.  Only MDR analyses was performed on lymph node aspirates.  No further analyses were performed.

Lines 182-183: Why were dogs 5 and 6 included in the study if not treated with metformin? 

While these dogs were initially identified as potential candidates for the study, owners elected not to participate further due to time and cost constraints. 

Table 2: not found. Please check the manuscript and include table 2

Table 2 has been added. We apologize for its omission. It was an oversight on our behalf.

Reviewer 2 Report

This is a very interesting and well drafted manuscript that presents findings of a study that evaluated changes in mRNA expression of MDR markers and a complete tumor response following treatment of dogs with MDR B-cell lymphoma with metformin.  Metformin reduced markers of MDR in 4 dogs, and 1 dog had an objective response to treatment with metformin + CHOP.  Interrogation of samples collected from dogs with MDR B-cell lymphoma implicated impaired APC function and not ABC transporters in MDR.  Investigation in MDR resistant canine lymphoma cells revealed that exogenous activation of APC could restore chemosensitivity.

I think these results offer interesting preliminary insight into a pathway that could be targeted in MDR cancer.  It would be interesting to see if this pattern of APC inactivation is repeatable in a larger sample of dogs with MDR B-cell lymphoma (or T-cell lymphoma, which tends to be associated with reduced response and shorter survival compared to B-cell lymphoma), and I hope that the authors are pursuing a prospective evaluation in which dogs are randomized to receive metformin + CHOP or placebo + CHOP.  There are conflicting reports in human oncology (e.g., Cancer Metab. 2020 Jul 6;8:10. doi: 10.1186/s40170-020-00213-w.; Br J Haematol. 2019 Sep;186(6):820-828. doi: 10.1111/bjh.15997. Epub 2019 May 28.; Leuk Lymphoma. 2017 May;58(5):1130-1134. doi: 10.1080/10428194.2016.1239822. Epub 2016 Oct 5) on whether metformin improves outcomes or not in lymphoma, but I did not see references to any of this or other clinical applications in human oncology mentioned, which may be of benefit to the reader.

Page 1, line 39 – consider: “…have contributed to” improved “survival and disease-free duration”

Page 2, methods – Did the dogs that were enrolled have a confirmed diagnosis of B-cell lymphoma based on histopathology and immunohistochemistry, or was the diagnosis established based on cytology, immunocytochemistry, flow cytometry or PCR for antigen receptor rearrangements?

Tables – I do not see a “Table 1” or "Table 2" included in what was sent for review.

Page 4, line 197 – I think the authors mean “remarkably,” but I would likely omit either word and just present the results.

Author Response

Reviewer 2

Reviewer 2 is thanked for the insightful comments on our manuscript.

This is a very interesting and well drafted manuscript that presents findings of a study that evaluated changes in mRNA expression of MDR markers and a complete tumor response following treatment of dogs with MDR B-cell lymphoma with metformin.  Metformin reduced markers of MDR in 4 dogs, and 1 dog had an objective response to treatment with metformin + CHOP.  Interrogation of samples collected from dogs with MDR B-cell lymphoma implicated impaired APC function and not ABC transporters in MDR.  Investigation in MDR resistant canine lymphoma cells revealed that exogenous activation of APC could restore chemosensitivity.

I think these results offer interesting preliminary insight into a pathway that could be targeted in MDR cancer.  It would be interesting to see if this pattern of APC inactivation is repeatable in a larger sample of dogs with MDR B-cell lymphoma (or T-cell lymphoma, which tends to be associated with reduced response and shorter survival compared to B-cell lymphoma), and I hope that the authors are pursuing a prospective evaluation in which dogs are randomized to receive metformin + CHOP or placebo + CHOP.  There are conflicting reports in human oncology (e.g., Cancer Metab. 2020 Jul 6;8:10. doi: 10.1186/s40170-020-00213-w.; Br J Haematol. 2019 Sep;186(6):820-828. doi: 10.1111/bjh.15997. Epub 2019 May 28.; Leuk Lymphoma. 2017 May;58(5):1130-1134. doi: 10.1080/10428194.2016.1239822. Epub 2016 Oct 5) on whether metformin improves outcomes or not in lymphoma, but I did not see references to any of this or other clinical applications in human oncology mentioned, which may be of benefit to the reader.

It is true that not every case of metformin treatment results in improved cancer outcomes, but the bulk of the meta-analyses conducted do show a positive response. I included the first reference but not the next 2. I have softened the text on current line 71 to read: …lower incidence of many cancers compared…

Page 1, line 39 – consider: “…have contributed to” improved “survival and disease-free duration”

Done. Added.

Page 2, methods – Did the dogs that were enrolled have a confirmed diagnosis of B-cell lymphoma based on histopathology and immunohistochemistry, or was the diagnosis established based on cytology, immunocytochemistry, flow cytometry or PCR for antigen receptor rearrangements?

B-cell lymphoma was confirmed through histopathology and IHC. Initial diagnosis was based on cytology.

Tables – I do not see a “Table 1” or "Table 2" included in what was sent for review.

This has been added. Sorry for the confusion.

Page 4, line 197 – I think the authors mean “remarkably,” but I would likely omit either word and just present the results.

Done, removed the word “remarkedly”, which should have been “remarkably”. Neither words are currently present.

Round 2

Reviewer 1 Report

The authors responded to all my comments and suggestions. I find the new version clearer than the previous one. I cannot find tables 1 and 2 also in this version. Considering that the Authors reported the inclusion of the tables, I will ask the Editor to check it.